# Importance of STEM and STEAM Education for Improvement of the Land in the RURAL Environment: Examples in Latin America

Elisa Gavari-Starkie [1], Patricia-Teresa Espinosa-Gutiérrez [2,3,*], Cristina Lucini-Baquero [3] and Josep Pastrana-Huguet [2,4]

[1] Department of History of Education and Comparative Education, National University of Distance Education (UNED), 28040 Madrid, Spain; egavari@edu.uned.es

[2] International Doctoral School of the UNED (EIDUNED), National Distance Education University (UNED), 28040 Madrid, Spain; joseppastrana@msn.com

[3] Plant Production and Agrifood Quality Research Group (PROVECAv), St. Teresa of Jesus Catholic University of Avila (UCAV), 05005 Avila, Spain; cristina.lucini@ucavila.es

[4] Consell Insular de Menorca, Balearic Islands, 07702 Menorca, Spain

* Correspondence: pt.espinosagutierrez@gmail.com

**Abstract:** Interdisciplinary STEM and STEAM education gives us the opportunity for training students to achieve all educational objectives in a sustainable development framework. Land in a rural area is a scenario with a range of educational resources for the development of STEM and STEAM projects to train students to interact with the rural environment. The possibilities of the land in order to prepare students for the needs of today's world are linked and sensitized to the environment. In this article, we will offer contemporary examples of STEM and STEAM projects that have been carried out in Latin America. These projects are being developed in which young people establish links with their environment, territory, and the local environment surrounding them. We must look carefully at projects and works in which students' ties to their land and environment are valued through the STEM and STEAM education necessary today. In this scenario, a comparison has been made between the projects in Latin American countries and Spain.

**Keywords:** STEM and STEAM education; interdisciplinary education; sustainable development; environment; rural environment; land; Latin American; Spain

## 1. Introduction

Rural environments worldwide offer endless educational resources that must be valued. While it is true that countries in different parts of the world have their own rural situations and therefore precise comparisons cannot be made, it is necessary to look at what some countries are doing to educate and create a population that sustains the rural environment and the planet. The education one receives throughout one's life and especially from an early age will constitute one's outcome as a human being. According to Luengo [1], the first maternal care, the social relationships that occur within the family or with groups of friends, attendance at school, etc., are educational experiences that shape our way of being.

The importance of the concept of education is complex. What precisely does education mean? School, high school, and university are linked closely for most people. However, it must be agreed that education is a broad concept closely related to the experiences of human beings throughout their lives.

We must ask ourselves what the purpose of education is, why one has to be educated, and what one is looking for when one is looking to be educated. As Touriñán exposes [2], the fundamental objective of education, as a result, is the acquisition in the educational process of a set of behaviors that enable the student to choose, commit, decide, and carry

out their personal life project. These behaviors will constitute a means to create valuable people and are a form of insertion and prosperity in human beings.

There are different types of education; one could speak of formal, informal, and non-formal education, although according to Pastor [3], the concepts of formal and non-formal education have an evident historical and political relativity, since an educational activity can be formal in a country and non-formal in another. At this point, we can reflect again since, in each country or territory, a specific type of education will be binding to the formation of their citizens, and this will be so for different reasons, mainly social reasons.

As Delors affirms [4], one of the four learnings around education throughout life will be "learning to do" to influence one's environment. Training people to impact the environment positively is a current objective in education.

Education must generate a path of improvement, and in our current world of the 21st century, our societies must be able to commit to the improvement of the planet, which means that individuals must be involved in caring for their environment and habitat.

Faced with environmental problems such as those arising from global climate change and what this implies, such as droughts, fires, reduction in ice at the poles and other catastrophes, a possible solution that involves human beings in respecting the environment must be sought through education. According to Hall and Bridgewater [5], the time has come for approaches to mitigate environmental deterioration to be given priority in the context of communication, education, and public awareness. Otherwise, we risk entering a spiral of environmental destruction with dire consequences for humanity.

With the emergence of the concept of environmental education in the 1970s, at the International Environmental Conference in Stockholm, education has contributed to generating changes in people, their attitudes, and their environmental values. Environmental education and the development of projects from school linked to respect for the Earth train children, who will be our future society, to be more sensitized from an early age to their urban or rural environment. Environmental education constitutes the educational process that deals with the relationship of the human being with their environment (natural and artificial) and with themself, as well as the consequences of this relationship [6] in the human being themself, in their land, their environment, and obviously in their life.

To review the origins of when environmental education became important, one has to go back to the preface in the book edited by UNESCO in 1980 [7], which includes the main guidelines of the UNESCO Tbilisi Conference. The Tbilisi Conference [8] constituted the starting point of an international environmental education program, according to the unanimous desire of the Member States. Environmental education began to gain strength precisely at that moment, a decade in which the planet was already experiencing immense noticeable environmental changes due, among other causes, to significant industrial and technological advances. In that same book, it is stated that environmental education, properly understood, should constitute a general continuing education that will react to the changes that occur in a rapidly evolving world [7]. That leads us to think about the importance of a permanent education that can and should be developed to achieve the union of the human being with their environment, with the environment that surrounds them, with the land that surrounds them, always taking care of nature and in short, taking care of the planet. The same document says that by adopting a global approach rooted in a broad interdisciplinary base, environmental education creates a general perspective within which a profound interdependence between natural and artificial environments is recognized. That leads us to think about the importance of interdisciplinary education today since our current world is interdisciplinary, and interdependence between the natural and artificial environment exists today. As humans, we must take advantage of resources that can provide interdisciplinary education, improve human beings, and make people more sensitive to caring for the environment, land, and planet.

As Labrador and Del Valle [9] mention about the Moscow Congress of 1987 [10], environmental education and the perspective that it should be addressed to the public of all ages and educational levels, also in non-formal education and education of adults, for

specific, very diverse, professional and/or social groups. Within this diversity, including people who live in rural areas is necessary and very valuable. Environmental education also harbors the nuance of transdisciplinary. As Schrodinger [11] suggests, transdisciplinary is one of the highest ideals of environmental education, in that environmental objectives can only be achieved by addressing the issues across all areas and forming new associations. Martínez [12] explains that there has been awareness of the interdependence between the environment, development, and education based on a transversal and transdisciplinary response providing an educational paradigm. Working on environmental education is the basis for generating a series of changes in the behavior of human beings that will lead to creating more just, more developed societies that are more careful with their habitat, which is the land on which they live, their environment, and the planet.

Considering everything mentioned, education for sustainable development has been proposed for some years. Currently, the 17 Sustainable Development Goals (SDGs) of Agenda 2030 [13] will lead us to a better and fairer world. Goal 4, which proposes quality education, is essential to educate our children.

A permanent, interdisciplinary, and transdisciplinary education will train citizens to be better prepared for the world they have had to live in; a world of problems and advances. When talking about the problems, it is necessary to recognize the social, economic, and environmental issues, territories with hunger and wars, which humans must be capable of improving in many aspects to affront those. An education that trains citizens with interdisciplinary and transdisciplinary approaches will help people emerge better developed to adapt to these times. Permanent education should also be designed to sensitize these people from an early age to their land, environment, and planet. In that case, people will be better formed to live on this planet through knowledge and thus be more sustainable, resilient, resolute, and committed to their environment.

STEM education was born in the United States in the 1990s, as the integration of sciences, technology, engineering and mathematics. STEM is an acronym that serves to refer to the professional field that includes the different scientific–technological disciplines but also to refer to the set of knowledge, skills and practices that must be promoted and developed throughout schooling [14]. Educational policies in different countries are recognizing the importance of STEM education [15].

The STEM educational model has, as its pedagogical model, the promotion of academic autonomy for students through the development of critical thinking in environments oriented towards the formation of learning communities inside and outside of the classroom. In 2006, the STEM model was redefined, incorporating art into its structure, and called STEAM [16]. This educational model allows artistic skills such as empathy or creativity to address science, technology, engineering, and mathematics from different perspectives. A collaborative learning approach transforms theories into models applied to real life and the world around them. It is a strategy to know and understand reality throughout life from inquiry and interaction.

Interdisciplinary STEM and STEAM education presents an opportunity for creativity and educational innovation in a world scenario that needs training in which students can achieve all educational objectives and are prepared to achieve sustainable development.

The STEM Women's Congress held in 2021 discussed how science unites with the environment through STEM education [17]. According to Pearson group [18], STEM projects with a focus on social responsibility are ideal for raising awareness and instilling values in the environment since they start from real needs (for example, using greener energy or improving the nutrition of a community) and empower students to find solutions on their own. STEAM methodology involve more than the incorporation of the arts; Piaget and Vygotsky have already pointed out the processes of assimilation, accommodation, scaffolding, and a global vision of education is the novelty of this methodology, as it does not work with science, technology or the arts independently [19]. According to the OAS practical guide, to achieve an inclusive society, it is essential to incorporate a gender perspective in STEAM education programs and initiatives, and STEAM professions are rapidly growing

and have greater influence in promoting innovation and the economic development of countries. STEAM education also promotes individual and social–economic success [20]. As Jiménez [21] states, according to data from the Ibero-American Science Observatory, Technology and Society (OIE), 2020, educational management strategies must be reoriented to guarantee the increase in scientists with interests in the formation of identities in science, technology, engineering, arts and mathematics, addressing global and national problems of Latin America and Caribbean countries.

Educating for the environment is necessary; unfortunately, it is often not a current goal. This article highlights the importance of global action to make environmental education a necessary reality for sustaining the planet. The 2030 Agenda for Development Sustainable highlights the importance of the STEM approach to education development. It refers to the relevance of the participation of women in STI through STEM. Furthermore, this entity displays the "education in science, technology, engineering, and mathematics (STEM); and education for sustainable development (SDG) as part of a quality education" [13]. Vallejos and Callao [22] emphasize the need for the educational and environmental process to lead individuals to perceive themselves as members of and as co-responsible for the environment; however, teachers, scientists and public policy makers still have a limited vision. There are countries like Colombia where the concepts and characteristics of environmental education clearly stated in the policy of education, environmental education and the PRAE are the pedagogical strategy for the inclusion of environmental education as a transversal axis in the educational formal system [23].

Currently, in order to improve the current rural situation, STEM and STEAM education is being developed in Latin American countries (such as Colombia, Chile, and Peru) and in European countries (such as Spain), with significant differences, but with a shared background towards improving the rural situation.

The rural women in the Latin American countries studied in this article suffer from a gender gap that translates into the early leaving of school, with all that this fact implies. Carrying out STEM and STEAM education to address the gender gap is important nowadays and even more so in rural areas. Many countries develop programs along these lines.

Local sensitivity is achieved by valuing and using the resources of the rural environment to carry out STEM/STEAM education. Activities, sessions, and projects are developed that link students with their environment, their habitat, their people, their culture, and their traditions; with all this, we manage to enhance sensitivity to the local communities, which will result in a desire to stay in the countryside, caring for it, respecting it and working sustainably with everything that the countryside has to offer, and of course, their land. It is essential to set up governmental educational programs that work on the rural environment and gender.

According to Rubio [24], the depopulation and ageing of rural communities are problems that occur equally in Europe and Latin America; they are caused by the lack of generational replacement in the activities that persist, and the slower economic development of the latter aggravates poverty. Land in rural areas is fundamental: it provides food, income from its cultivation, and economic and social possibilities, thus its care and development are essential to rural areas. Rural dwellers have been tied to their land for generations and their care, knowledge, and maintenance keeps the rural area and its people alive.

## 2. Objectives and Methods

This study aims to demonstrate that STEM/STEAM education applied in the rural world can be an integrative tool that serves as an impulse for the creation of new citizenships committed to social and environmental justice and, at the same time, are committed to achieving changes that lead to creating links with the territory in which they live and increasing opportunities in this environment. This article is a source to see the rural world and its resources as an educational source to carry out STEM and STEAM education in students, observing STEM and STEAM education as an education capable of training students who can achieve all educational objectives and are prepared to achieve sustainable

development. To this end, and in order to narrow down our research, some experiences of STEM/STEAM education to improve skills in rural areas of Latin American countries are fundamentally examined. We mainly focus on Colombia, Chile, and Peru countries, where interesting STEM and STEAM education is developed.

The specific actions developed in this study analyzed the situation in Latin American countries, especially Colombia, Chile, and Peru, regarding education in rural areas and STEM and STEAM education. Likewise, we proceed with are view of educational programs and laws developed in these countries, collecting successful cases of the implementation of STEM and STEAM education in rural areas.

In addition, various actions of Spain in this area are reflected, not to make a precise comparison with Latin American countries, since this is not possible because they are countries with very different situations, but to highlight the advances in STEM and STEAM education for the improvement of education in the rural world in different Ibero-American countries.

This article reflects comparative educational research using qualitative methodology. The qualitative study is based on a bibliographic review of documentary sources on the evolution of STEM and STEAM, focused on the rural world. Considering the research, the selection of primary and secondary sources was made considering the following keywords: STEM and STEAM; Latin America; rural education; Colombia; Chile; Peru and Spain, since 2000.

The results were systematized and organized to facilitate this study. The primary sources are those based on reports, legislation, and other institutional publications of governments and international organizations such as the United Nations. Regarding secondary sources, a search was carried out for scientific research articles related to the topic of study in bibliographic databases and storage systems for academic publications. All of this information has been categorized into three levels of information, considering the theoretical–conceptual approaches provided by various researchers, institutional approaches and documents, and some contributions from non-scientific publications associated with the object of study.

### 3. Education for the Improvement of the Land in the Rural Environment

Currently, the rural population represents 70% of the world's population, and inequalities between the countryside and the city constitute a significant obstacle to sustainable development. This is why rural education is traditionally addressed with subsidiary policies, especially in developed countries [25].

In the midst of the COVID-19 pandemic, villages became important safe havens where many people from the cities went to live and work. Before the pandemic, many urbanites from the cities only saw towns as a place of rest, but towns should be a place to live in an excellent way with all the benefits offered by rural areas at the service of citizens. As Hernández [26] points out, it is important to highlight the recovery of cultural heritage for the population and the projection of local culture for society as a whole. The territory must be presented as an opportunity to live. However, these roots are maintained, and with this, there is an improvement in the rural environment and its land and people, which is a direct consequence of our fundamental factors to fix the population in rural areas through education, health, technology, and employment [27]. Gavari, Espinosa and Lucini [27] point out that education is the protagonist, and the other three factors revolve around it. Let us then recall the 17 SDGs of the 2030 Agenda, which revolve around SDG 4; women being employed is of vital importance, hence the importance of gender equality in terms of educational and employment opportunities in rural areas.

It must be noted that a rural environment can show many problems and under recognized values. As Amiguinho [28] explains, emphasis must be placed on participatory insertion in the closest community to be a citizen of an increasingly globalized world: a citizen of the world is true from somewhere, with roots in the "local". These roots are maintained, and with this, there is an improvement in the rural environment and its land

and people, which will be a direct consequence of the four pillars that must be assembled for functioning in the rural environment: education, technology, work, and gender.

The rural environment offers us agricultural and forestry resources that must be exploited, and to do so, we must think about the fundamental physical, biological, historical–cultural, and economic values that are usually minorly recognized in the rural environment such as food production, rural environment reserves like minerals, plants, and animals, the rural environment presenting an oxygen factory at the service of humanity thanks to its forests, the rural environment as a source of culture in the form of traditions, customs, cuisine, and others, and it being a source of architectural resources, archaeological resources, and landscapes that are spaces of incalculable value [29].

In this sense, educational services for the improvement of rural areas stands out. How? By forming human beings that are better linked to their environment, their land, and their habitat. Thus, education can form human beings linked to the land they live on. But let us not only think of the youngest and of our future society, but also of our elders in the rural population.

Undoubtedly, citizens' education achieves economic, social, and environmental development. Economical because the best-educated citizens are able to face the financial problems that arise in their lives and their work; social because education makes human beings grow in all their social landscapes; and environmental responses because it can develop better-trained citizens in this area.

To the permanence of the rural environment, it is crucial that people can live in towns because there are means, work, and economic resources for their population. It is also vital that rural inhabitants have their full social expectations. To this end, it is important to address the problems of depopulation, aging, and gender in many rural areas. In the villages, we need to value our elders and women because if there are no women, the population becomes masculinized, and the village dies. The environmental ties from the rural world to its inhabitants are also crucial for maintaining the rural environment. If its inhabitants are sensitive to their environment and land, they will want to stay in it and live there, lasting with it. In this way, they will try to stay in rural areas, on their land.

In short, the importance of education, particularly STEM and STEAM education, to develop a current, interdisciplinary, transdisciplinary, and lifelong vision of the economy, society, and the environment using the educational resources offered by rural areas should be highlighted. The development of STEM and STEAM education projects in this area will train people who are better prepared in rural areas and are more linked to their land, with all that this implies. The rural environment with its land is a scenario that offers us a range of educational resources, which must be used in schools and institutes, linked to the agricultural and forestry world and also social, to, among other aspects, propose the development of STEM and STEAM projects in which students who are better trained and prepared for the needs developed of today's world, as well as students who are more linked and sensitized to the environment, to the rural environment and its land, thus are important in this time. With these resources, rural students can and should achieve interdisciplinary knowledge in science, technology, engineering, arts, and mathematics and be more prepared for today's world, since one of the objectives of this methodology is to contribute to developing student competencies. The 21st century has developed such that more children can become leaders, innovators, and researchers, and develop the tools necessary to face today's challenges and their communities' future [30].

The rural environment has social resources that should be paid attention to. A rural environment that suffers from depopulation, with a large part of the population aged in many areas of the planet, must take advantage of the knowledge and wisdom of its elders as educational resources. With their experience, culture, and wisdom, older people can become an additional educational resource, transmitting their knowledge to new rural generations [31]. Also, STEM and STEAM education can generate knowledge in reverse, generating new knowledge in older adults, such as the European Commission's "BRAIN" project, which aims to transmit a substantial and sustainable impact; older adults involved

in the local program, after having acquired new knowledge and skills, will act as replicators of their learning that will raise their self-esteem and promote their social inclusion [32]. This project, called "BRinging STEM into Active AgINg", has been financed with the support of the European Commission, and its information is included in the framework of the European program "Erasmus Plus KA2 Strategic Partnerships for Innovation and the Exchange of Good Practices".

**4. Rural Situation in Latin America**

Although it is true that already at the beginning of the 21st century, as Hendel affirms [33], several Latin American countries, mainly Bolivia, have begun to experience profound political transitions that propose the redefinition of the organizational parameters of their economy, their politics and their state, outlining a questioning of the notion of development that characterized the rural development policies of the world's great powersduring the 20th century.

Talking about the rural situation in Latin America is complex due to the number of countries we are talking about; all of them have differences, but they also have common points, such as the spoken language. The definition of urban and rural space in most countries is complex because the meaning of one does not default with the other. However, similar quantitative and/or qualitative criteria are generally adopted [34].

Poverty is one of the most persistent features of Latin American society and has been resistant to the conventional policies designed to reduce or eliminate it [35]. According to Clausen [36], rural territories are, therefore, still far from meeting the objective of not being "left behind" in eradicating all forms of poverty.

Bolivia, Chile, Colombia, Costa Rica, Ecuador, El Salvador, Mexico, Paraguay, Peru, and the Dominican Republic are countries that have unquestionable natural and cultural wealth. Approximately 20% of the world's oil reserves, 25% of strategic metals, and more than 30% of primary forests are in the Latin American and Caribbean region [37]. The recognition of the use of the natural landscape to achieve a better quality of life is an element that has gained validity from rediscovering the concepts of rural development and new rurality [38]. Exploiting natural resources sparks disputes about the use of water, biodiversity, land, and critical ecosystems, among others, and about the repercussions that exploitation has on them [37].

It is important to know the situation of each rural territory, as well as the problems of the people who live there, especially the most vulnerable people: people of different ages (children and older people) and women in rural areas. The Latin American Center for Rural Development (RIMISP), a network formed in 1986 by different partners and allies, helps to understand rural transformations and formulate better strategies to achieve equitable territorial development in Latin America [39]. In the rural environment, understanding the rural situation, the environment, the people who make up it, and their problems, as well as moving towards the development of the land, is vital. The RIMISP highlights the importance of proposing respectful, relevant, and innovative research strategies that involve the voices and knowledge of territorial actors, considering ethnic, generational, and gender diversity [39].

*Rural Situation in Colombia, Peru, and Chile*

Colombia is the second most populous country in South America and the fifth largest in territory in both Latin America and the OECD [40]. In Colombia, the economy has recovered significantly from the COVID-19 crisis, and a strong response in monetary and fiscal policies has managed to avoid a further contraction in income. Economic growth has been observed, but to mobilize rural potential, the government must prioritize the structural challenges historically limiting rural development, such as gaps in access to quality health and education services [40]. Actually, Colombian rural communities are experiencing profound transformations in public policy. However, the institutional framework still maintains a rural development bias focused on primary activities, social

assistance, and security as a legacy of a historical vision of development focused on urban [40]. Deforestation has increased in Colombia, and to achieve current greenhouse gas emission reduction targets, further reductions in deforestation will be necessary, with the country committing to reducing net deforestation to zero by 2030 [40]. According to the National Administrative Department of Statistics of Colombia (DANE) [41], in Colombia, the unemployment rate throughout 2021 and half of 2022 has been higher for rural women than for men, with a rate gap between 6.8 and 9.7%. The population in rural regions in 2021 was higher than the average of other Latin American OECD countries (Mexico and Chile) [40]. In 2022, 23.7% of the population, 12.2 million people, were located in rural areas, and among the rural population, 48.2% are women and 51.8% are men. On the national average, women are 51.2% [30]. Contrary to the global aging trend, Colombian rural regions benefit from a high young population (26% in 2021), well above the OECD average (17%), and other Latin American OECD countries, such as Chile (19%) [40]. This fact is perfect for the rural world and its development, since rural areas will remain associated with technology, innovation, and the economy. Colombian rural regions are a potential source of wealth and wellbeing and contain one of the most diverse ethnic presences in South America and a more significant number of young population than the OECD average [41].

Peru is the seventh American country in terms of population. In 2022, its population was 3,396,700, 50.4% women and 49.6% men, settled in 1874 districts [42]. In recent years, inequality of opportunities between rural and urban areas in Peru has been pronounced. Following the recession caused by the COVID-19 pandemic, the economy rebounded quickly but has since slowed sharply to just 1.1%. It is expected to increase gradually by up to 2.7% in 2024 [43]. Looking back and seeing how its population has changed from 2016 to the present, we can observe small changes. In 2016, 20.8% of the population in Peru lived in rural areas, and only 28.2% accessed all drinking water, electricity, and sanitation services vs. 82.6% of urban residents [44]. According to the latest national survey on perceptions of inequalities (ENADES 2022), 61% of the people surveyed perceive a severe disparity between Peruvians living in cities and rural areas [45].

Chile is the ninth Latin American country in terms of population. According to data from the National Institute of Statistics of Chile (INE) from the end of 2021, the population figure is 19,458,000 people [46]. The Chilean economy recovered quickly from the pandemic, and inflation has risen to a 30year high, with peaks close to 14% annually in 2022, driven by an expansionary fiscal policy and exacerbated by global supply constraints and the war in Ukraine [47]. Since the population and housing census carried out in 2017 [48], it can be seen that its population has increased until now, since it was a total of 17,574,003 people, of which 8,601,989 (48.9%) are men and 8,972,014 (51.1%), women [46]. Currently, 25% of the Chilean population live in territories (82% of the total territory) with high rurality. Environmental challenges and risks are important, but they also offer significant opportunities for the future [47]. Rural–urban migrations, the productive reconversion of farms that have reduced their workforce, and earthquakes have produced unemployment in the countryside, which goes hand in hand with the de-agrarianization of local activities [49].

## 5. State of the Art of Rural Education in Latin America

At this point, the characteristics of the education received in rural schools are noted. According to Caliva [50], the contents are conceived in most cases from situations outside the rural area and without sufficient consultation, thus resulting in universalizing, academic, monotonous, excessively extensive curricula with very little range for adaptation to the local situations. This is something to reflect on since, in an environment so rich in land resources and landscapes, using curricula that are not adapted to the environment of the rural situation from an educational point of view has its consequences. Education must be capable of responding to each territory's needs. Emphasis should be placed on the need to promote innovative educational processes based on the needs and potential of the different rural populations of Latin America [50]. As cited by Arias [51], today's rural schools must

teach school content and recognize local knowledge related to agricultural work, roots in the land, environmental sustainability, and community history.

The educational policy of most of the Latin American and Caribbean region began to experience a reactivation at the end of the 20th century and, especially at the beginning of the 21st century [52]. In 2023, the Regional Educational Policy Forum was framed in the political implications and implementation strategies of SDG 4 of the 2030 Agenda [53]. Working on this SDG is necessary in all areas of the world. Still, in this Forum, the idea was to think about how digital technologies can contribute to transforming the educational systems of Latin America and the Caribbean [53].

Illiteracy among adults and youth continues to be a mainly central to rural populations, and the preschool level practically does not exist in these areas [54]. However, we must highlight that rural education in Latin America is a viable possibility [55]. According to Santa María et al. [56], researchers agree that to confront the problem of educational inequality, a drive for change is required that addresses the factors that mark said inequality.

Latin American countries that achieved coverage of the school system present serious quality problems in the learning results of boys and girls who participate in learning achievement measurements [54]. At international events on education, proposed agreements are made to improve the population's learning levels and prepare students for life [56]. The latter is essential because educating the population about rural life prepares rural inhabitants to live harmoniously in their local environment, consequently improving their quality of life.

The International Congress Epistemologies of the South and Latin American Ruralities held in 2019 focused on rural education, the pedagogy of Mother Earth and the ecology of knowledge and territory, women and Latin American spiritualities, ancestral practices, and knowledge [57]. It is worth highlighting the importance of the Ibero-American Colloquia on Rural Education that the Thematic Network for Research in Rural Education (RIER) [58] has been developing for some years now in rural schools. The VI Ibero-American Colloquium on Rural Education was the most recent, held in July 2023 and developed by the Rural Education Division of the National University of Costa Rica, in collaboration with RIER.

*Rural Education in Colombia, Peru, and Chile*

In Colombia, school dropouts have increased considerably due to the pandemic, especially among students from disadvantaged socioeconomic backgrounds [59]. According to data collected in DANE [41], in Colombia in 2021, 75.4% of rural women and 75.2% of rural men between 6 and 21 years old were studying, while in urban areas, these percentages are 79.3% and 79.2%, respectively. Women suffered from dropping out of school due to home and family issues. 11.2% of rural women between 6 and 21 years old who do not study do so because they must take care of household chores, and 4.4% do so because of pregnancy [41]. Dropout rates in secondary education, which tend to focus on students from disadvantaged socioeconomic backgrounds, increased in 2020, and only 50% of children aged 3 to 5 years have access to preschool education [59]. In the General Education Law of 1994 [60], article 10 describes formal education, article 36 refers to non-formal education, and article 64 refers to peasant and rural education [61], mentioning that the National Government and the territorial entities will promote a peasant and rural education service, which will mainly include technical training in agricultural, livestock, fishing, forestry, and agro-industrial activities that improve human conditions, work, farmers quality of life, and increase food production in the country (art. 64) [60]. In this country, the PER Rural Education Program is currently being developed. It has Phases I and II, implemented since 2009, to mitigate the problems affecting educational coverage and quality in rural areas, helping to overcome the existing gap, and the Special Rural Education Plan [62,63]. In Colombia, there is a clear intention to rescue the living memory of past adversities, recognize forgotten population sectors such as farmers, Afro-descendants, and displaced people, and defend the territory [57]. According to the data provided by the ICFES and the OECD, it is evident the challenge rural education represents for the country

in its commitment to improving educational quality [64]. It is also worth remembering the V International Congress of Local Development held in 2019, whose motto was Socioeconomic Systems with Territorial Anchors, where it reflected how local areas should be promoted as places to satisfy economic, social, cultural, and environmental needs [65].

In Peru, public spending on education is low (2.7% of GDP in 2018). At least 96% of adults completed primary education in 2020, thus surpassing other Latin American countries [43]. In 2018, 24% of secondary school students were behind in school versus 7.4% in urban areas, and 8.7% of women over 15 years of age and older versus 3% of men are illiterate, according to data from CREER (growing with multigrade rural schools in Peru) [66]. This information about women's education is significant. The school dropout rate in rural areas is a fact, and in rural areas, women represent the population with the highest dropout rate (8.6%) compared to men (6%). In rural areas, only 32% of the population completes secondary education, and 8% ends higher education [43]. In secondary school, the gender gap remains relatively high, with 34% of women over 15 years old, compared to 42% of men, completing secondary education [67]. A higher proportion of 15-year-old girls achieve below-average results in mathematics and science compared to their male peers, while girls achieve better results than boys in reading [43]. The work obligations of adolescent girls outside of school and teenage pregnancies are the main factors that explain female-school dropouts, especially in rural areas [43]. There is a profound difference between urban and rural education, evidenced even more since 2020 from social isolation due to the COVID-19 health emergency [42]. Among the actions implemented in the country, we can highlight the Non-School Initial Education Program (PRONOEI). This public, early childhood education program comprehensively provides care and education to children in rural and remote areas with limited access to formal education [68]. And also, the UNESCO Horizons Program, a rural secondary education program that works in six regions of Peru (Amazonas, Ayacucho, Cusco, Piura, Arequipa, and Puno), ensures that adolescents in public schools located in rural territories finish their secondary school studies [69,70].

In the case of Chile, the school system is structured in two mandatory cycles: 8-year primary education (1st to 8th grade) and 4-year secondary education cycle (1st to 4th grade) [25]. The CASEN 2022 survey showed 62% more rural poverty than urban poverty in Chile, a figure also supported by the 57% corresponding to extreme poverty rates [71]. Several laws and programs must be highlighted in Chile concerning rural education: The Rural Basic Execution Program of 1980, the Quality Improvement and Equity Program (MECE/Basic/Rural) that took place from the 90s to 2002, Decree No. 4 that regulates the Rural Basic Education Program of 2001, and Decree No. 968 of 2012 [25]. In 2020, 52 proposals were proposed to improve rural education in Chile: institutionality and public policies for quality rural education, curriculum and teaching roles with a territorial perspective, educational offer for a lifelong trajectory, and infrastructure. And schools that favor learning objectives were introduced [72]. In Chile (2019), there are 3401 rural schools, and around 1800 correspond to multigrade establishments; that is, they work with students from different grades in the same classroom [73]. Currently, female participation in the labor market is low, and expanding access to quality early childhood education would close critical gaps in cognitive and social progress and enable more women to work [47].

## 6. Results

### 6.1. STEM and STEAM Education in Latin America for the Improvement of the Rural Environment

According to Bonilla et al. [74], STEM education is an emerging trend that seeks modular, adaptable, and easy-to-use tools to help students in practical teaching. STEM/STEAM education offers economic, social, and environmental benefits to the rural population, among other things, by working interdisciplinary with science, technology, engineering, arts, and mathematics.

Working on STEM and STEAM education in rural areas of Latin America implies an improvement in the rural environment since participants in this type of education will

be better prepared to face the rural world in which they live their daily lives. Working on gender in rural areas, thanks to STEM/STEAM education, improves the environment in which such education is developed, in this case, the rural environment, since this implies better trained and prepared women. Better-trained people will continue to live in rural areas and take care of their homes, land, environment, and survival over time in that environment.

For the towns, it is vitally important that women remain in them. Of note in STEAM education, is the inclusion of discussion and analysis of gender inequality in the teaching and application of science and technology to encourage the participation of more women in these fields [75]. There are countries where being a woman continues to represent a severe disadvantage in science, both when choosing a career and when looking for a job related to disciplinary skills in STEM. As Arredondo et al. say, according to UNESCO, in terms of enrollment in STEM careers at the higher education level, women represent only 34% in Argentina, 25% in Chile, 30% in Brazil, and 38% in Mexico. In the case of Colombia, the global gender gap is 73% [76]. With the conviction that it is possible to break down gender prejudices and stereotypes, inspire girls in STEM, and increase female participation in this type of career, Ingeniosas: Science and Technology for All was born in Chile in 2016. Its initiatives have since expanded to Argentina and Colombia [77]. Considering the networks that promote the STEM/STEAM approach in Latin America, we can highlight the STEAM Laboratory in the rural classroom or the Educa STEAM Network [78,79]. The STEM Latin America Network, made up of more than 180 partner institutions from 14 countries, promoted by the Siemens Stiftung International Foundation, stands out. This network aims to form active citizenship that forges sustainable communities and territories [80].

STEM territories stand out as initiatives (46 already existing) that collectively promote STEM education in Latin America. Before creating a STEM territory, the actors promoting it are expected to identify territorial needs and/or problems, for which this structure could offer solutions based on educational innovation, community building, and sustainable territories. These initiatives are carried out in a total of eight countries. To begin, two questions need to be answered: Why promote a STEM territory initiative? What does the STEM territory initiative aim to promote (objectives)? [81].

The following table (Table 1) shows answers to these questions and also the union of all of this to promote the four fundamental axes already mentioned, so that the rural area lasts, which are: education, employment, health and technology.

### 6.2. STEM and STEAM Experiences in Rural Environments in Colombia, Peru, and Chile

In the case of Colombia, Cifuentes and Caplan [30] explain how, after the development of a successful experience of formal STEM/STEAM education in a rural school in Cundinamarca, STEAM education offers possibilities for raising the quality of education in rural contexts when focused towards a globalizing model in which diversity and inclusion are valued and respected. In Medellín, according to Cano et al. [82], implementing an integrated approach of STEM plus humanities translates into the best results and constitutes a fundamental factor in developing the local context. STEAM allows the generation of ideas to solve local problems and can contribute to developing interdisciplinary experiences, motivating students to participate in projects linked to the subjects that comprise it, and promoting cultural diversity [82]. According to Ochoa et al. [75], the STEAM community broadcasting solidarity extension project in Magdalena Medio was developed so that the peasantry can remain in the territory and live in harmony with animal and plant species through the defense of peasant culture and food sovereignty through agroecological practices, as well as peacefully resisting war [75].The STEAM project *Campesino* Laboratory for the Transition to Agroecology (*Lab Campesino*) [83], developed between 2018 and 2019, was a solidarity extension project led by groups of students from the faculties of engineering, sciences, and arts of the National University of Colombia and the rural organization, Tierra Libre, to generate a space for rural youth and children for exploration, experimentation and prototyping around agroecology, co-creation, and community organization, promoting

adaptive habitat, production, and consumption systems. Projects of this nature make rural inhabitants more linked to their environment and more respectful of the surrounding environment. Furthermore, according to Ochoa et al. [75], solidarity economy strategies are necessary to generate stable economic income to allow decent futures in the territory.

**Table 1.** Challenges of the Latin American STEM territory in conjunction with education, technology, health, and employment. Sources: www.educacion.stem.siemens-stiftung.org [81] and Gavari et al. (2022) [27] and own elaboration.

| Tracing Specialized or Government Information Sources | Recognition of Challenges that May Have Links to STEM Education | Some STEM Territory Objectives in Latin America | Relationship with Four Fundamental Factors to Fix the Population in Rural Areas |
|---|---|---|---|
| - Public politics<br>- Development plans<br>- Reports from observatories<br>- Local characterizations | - Sustainable development<br>- STI and citizenship promotion<br>- Learning gaps<br>- 21st century skills and problema solving | - Promote science, technology, research, and creativity, strengthen research processes, improve the quality of life of the community.<br>- Promote scientific vocations that seek social transformationto bridge the gap between men and women in STEM careers, integrating strategiclines in STEM Olympics, climate change, teacher training in STEM, project-based learning and challenge-based learning.<br>- Contribute to educational quality from the STEM approach to promote context ualized curricula, integrating relevant aspects such as climate change and sustainable development, teacher profesional development (training) and educational innovation. | - Education: With STEM education, quality education is developed for today's 21st century world.<br>- Technology: In STEM education, we work on, among other things, technology aimed at today's technological world.<br>- Health: We worked with STEM education to educate for climatechange and sustainable development, creating healthier and healthier environments. Fairersocieties are created by working on the gender gap.<br>- -Employment: Better-educated people are trained, prepared for the professiona ldemand of the 21st century. Work is being conducted to bridge the gender gap in the world of work in sciencecareers. |

In Peru, it is already expected that intervention projects will be increasingly based on systematic studies of schooling in rural areas and that public education will be developed with opportunity and relevance [58]. In 2017, a study carried out in the Peruvian Amazon explored the community practices and knowledge that Shipibo children manage, as intercultural actors, linking with extracurricular spaces where their life and activity take place: the house, the river, the forest, and the farm, giving meaning to the school space and taking ownership of the school [84]. The "VI National Educational Research Seminar" that took place in November 2018 in the city of Cusco was held simultaneously with the II Ibero-American Colloquium on Rural Education, addressing the topic of "Perspectives and challenges of rural education: dialogues between research and educational policy" [85]. To strengthen the STEM competencies of Peruvian teachers, the Pontifical Catholic University of Peru (PUCP) developed a free course to, among other things, analyze the role of education in forming citizens with local and global socio-environmental rights and duties [80].

In Chile, the Center for Didactic Research in Sciences and STEM Education (CIDSTEM) of the Pontifical Catholic University of Valparaíso developed a STEM education program in a "blended learning" format, creating 83 STEM educational resources with an inclusion and gender perspective [80]. Bascopé and Reiss [86] analyzed STEM education in Chile through the socio-ecological component, concluding that this approach, when applied to local challenges, opens up new sources of knowledge that are difficult to achieve with forms of learning, based on the transmission of knowledge. Faced with the global problem of climate change, the Research Center for Advanced Studies in Education, CIAE, the Climate and Resilience Center CR2 and Inquiry-Based Science Education, ECBI, of the University of Chile worked for the development and adaptation of educational resources in various formats on climate change, to design a certifiable course for teachers on the subject by surveying 125 teacher members of institutions linked to the STEM Latam Network were survey about their priorities [80].

### 6.3. Situation in Spain: Education for the Improvement of the Land in the Rural Environment and STEM and STEAM Education (Experiences in Rural Environments)

As indicated in Gavari, Espinosa, and Lucini [27], the problem of rural depopulation continues in Spain, affecting education and other sectors. This is because the small population in rural areas has aged and there are only a few young people trained in rural areas; the majority must migrate to urban areas to complete their training. This displacement is due, in part, to the gradual reduction in Spanish rural schools, which inevitably forces students to have to move to centers with a larger population.

Although there is a reduction in the population that corresponds to "emptied Spain" [87], we must highlight the recent concept of "new rurality" discussed by Trigueros et al. [88], where new functions are provided to the rural environment, such as recreational, residential, conservation, and protector of cultural and landscape heritage. In this search for the potential of rural areas and to improve them, the GEOVACUI project [89], developed by the Complutense University of Madrid, provides promising conclusions to change the view of the rural environment and promote it as a "living and active territory" [90]. It is possible that, if we combine the proposals obtained in said project, together with the possibility of accessing funds from the Recovery, Transformation, and Resilience Plan, we could speak of an excellent opportunity to promote the rural environment as an attractive environment in the 21st century [91]. In general, most of the actions undertaken related to the rural aspect, to date, are focused on women in rural areas, and it would be interesting to expand the focus to reach the entire educational community.

However, despite the key role that rural education plays in improving depopulated, rural Spain, it is only sometimes feasible to promote its improvement. The rural education sector involves several agents, and it needs open schools with teachers and students in person and available, with updated resources. It can be an engine of change towards the living and active rural territory. Compared to the urban school, the rural school presents some economic, geographical, and social peculiarities, such as multigrade classrooms, instability of the teaching staff, or geographical disaggregation. One of the most notable differences between rural and urban schools is the greater involvement of the student in the environment and its sustainability; therefore, promoting rural education under the scenario of the 2030 Agenda is a fact that will be more affordable in rural areas.

Regarding the quality of education, it is worth recalling the analysis of the PISA 2015 report by Domínguez and Sánchez [92], where the lack of public resources is already evident, which has implied the closure of rural schools. In the recent PISA 2022 report [93], the worst data has been shown in the education of Spanish adolescent students. However, we have found that the best results have occurred in communities with depopulation problems in rural areas, such as Asturias, Cantabria, and Castilla y León. This fact leads us to consider whether, despite the gradual reduction in the rural population, in those centers where rural schools are maintained, rural education is presented as a potential alongside urban education to promote the future of young people, stimulating their learning capacity and

interest in their education. It would be interesting to analyze the dropout rates in these same communities, even in rural areas, to be able to verify if, in these communities, captivating strategies and methodologies are developed from the point of view of educational success to extrapolate their models to other Spanish regions, where worse education data are recorded in the PISA report.

In 2022, the Ministry of Education, Vocational Training and Sports awarded aid worth one million euros to 30 projects developed to promote educational inclusion and innovation, including the project "The rural school as an inclusive model and element stimulator of culture and the development of the environment", developed at Grouped Rural School (CRA) Río Tajo (Alcolea de Tajo, Toledo), CRA La Coroña (Ceceda, Asturias), CRA De Lozoya (Lozoya, Madrid) and the project "Sustainability, health and sport in the rural environment", developed at CRA Bajo Gállego (Leciñena, Zaragoza) and Early Childhood and Primary Education center (CEIP) O Salvador (Pastoriza, Lugo) [94]. Seeing the rural environment as an opportunity to educate people more linked to their land and environment is noteworthy.

In Spain, the competency learning model on the current Organic Law of Education (LOMLOE) is based on the acquisition of eight competencies. One of them is Mathematics and Science and Technology (STEM) competence. In this way, the STEM educational approach is integrated into the competency-based curriculum by integrating mathematical and scientific, technology and engineering competence that entails understanding the world using scientific methods, mathematical thinking and representation, technology, and methods of engineering to transform the environment in a committed, responsible, and sustainable way [95]. The LOMLOE offers an opportunity to create an educational model based on the didactic capacity to train students in the competencies that enhance sustainability and resilient capacity [96].

Currently, the Rural Campus Program [97] is committed to the potential of the territory, the promotion of young employment, and the promotion of local talent, promoting the connection of the population from different areas with rural spaces, generating new forms of roots and ties, promoting and creating employment opportunities in the territory.

In Spain, some STEM/STEAM projects of great interest are being carried out in rural areas, such as the "Rural, Remote and Real, R3 Project" of the University of Deusto [98], to promote STEM vocations in rural schools through remote experimentation. The "STEAM Laboratory in the Rural Classroom" Project [99] aims to, among other things, promote the reduction in the gap between science and technology that affects this environment. The STEAM Alliance for female talent, Girls on the Foot of Science, is an initiative of the Ministry of Education and Vocational Training to promote STEAM vocations in girls and young people and reduce the gender gap [100]. It is developed in rural and urban environments. Still, it is obvious that in the rural environment, it takes on great significance, generated by the fact that there are women in the depopulated towns of this country, contributing in such away that constitutes the break of rural depopulation existing in Spain. Let us not forget that if towns become masculinized from a lack in female population due to unemployment, towns end up dying.

As Alba comments in an interview with Lacalle [101], it is important to facilitate access for boys and girls for STEAM training adapted to their socio-demographic reality and address the three factors that continue to hold back girls from becoming interested in technology: the messages they receive from society, gender stereotypes, and the lack of female role models in the technology sector. This responsibility, this sustainability, and this commitment are also the same in the rural environment when this competence is improved in the formal education of rural schools in Spain.

## 7. Discussion and Conclusions

This article shows the real and evident role in the time of STEM and STEAM education for rural environments; it promotes the birth of a rural society with social (with emphasis on gender, at this point) and environmental justice. STEM and STEAM education enhances

the creation of early sensitivities in students, developing emotional, sentimental, and sensitizing connections towards the rural environment and its territory. All of this shows us that STEM and STEAM education, carried out with these objectives in rural areas, favor the improvement of opportunities for its people.

The research carried out in this article aims to highlight the strengths of STEM and STEAM education for the improvement of rural environments, thus improving the lives of the people present there. This article is aimed at rural educators so that they see this type of education as an opportunity to improve the rural environment, its land, and its people. This extremely important work performed by teachers should not be forgotten; they have the duty to strengthen their didactic and pedagogical preparation, with the aim of facing new challenges in the cognitive development of students and applying learning strategies to allow a better understanding of the contents presented in the different subjects [22]. Rural areas need well-trained, up-to-date people and professionals prepared for an interdisciplinary world of work, and STEM and STEAM education develops people who are professionally prepared for the demands of today's work environment.

The rural situations of Latin American countries are different, and the rural situations between Latin America and Spain are also different. Although a precise comparison cannot be made because other countries have different educational models, it is possible to emphasize the advances in the countries studied to improve rural areas, their environment, their land, and, of course, the people who live there.

STEM education was born in the United States in the 1990s, and later, Yakman [102] restructured the term into STEAM, making citizens better prepared for the world we live in, which is an interdisciplinary world. With STEM/STEAM education, our children can be educated and prepared in the current knowledge of science, technology, arts, mathematics, and engineering in a united way and without barriers between disciplines; it is a holistic education for a holistic world.

This paper shows that STEM/STEAM education projects are carried out in all of the countries studied to improve the rural situation. Each country has different educational situations, so improvement is sought through different STEM and STEAM education strategies.

Women in rural areas in Latin American countries suffer from a gender gap, especially in the countries studied in this article. This leads to a gap and delays in the knowledge of the rural environment's inhabitants at different levels. For rural advancement to be as it deserves, it must be able to uplift women just as it does men, in full equality. Information has been collected on the educational situation of girls and women of rural areas in Latin America and Spain, and a series of STEM and STEAM projects of interest are being developed to fight against the gender gap.

Generating links with the environment and its land is allowed with STEM/STEAM education if the rural and forest resources of each rural area across the world are used, training people who are more prepared to know the land surrounding them. The rural environment and its people should be included in the world, as it is a source of wisdom, culture, and has resources of priceless value. Valuing the rural environment is vital in our time, to observe the territorial models of countries from other perspectives. The resources of rural areas are undeniable: landscapes, food, land, clothes, dances, wisdom of the elders, environment, etc. The rural environment offers a place to reside and must be cared for and maintained for future generations. Using the resources that rural areas provide makes citizens more linked and aware of the rural environment and land. In this way, children sensitized to rural areas will want to return to their land, live there, and work there, taking care of said land and keeping it alive without depopulation.

We observe that STEM and STEAM education in rural areas is necessary and possible, and even more enriching if the resources that rural areas offer are used as educational resources to their land and their environment, thus forming better-prepared citizens who are more linked to the land and its surroundings. Working on gender through STEM and STEAM education in different countries is necessary and even more so in rural areas, where it is observed that if villages become masculinized, they end up dying.

Latin America offers invaluable educational resources in rural areas to train citizens who are better prepared and more aware of their environment from an early age, forming citizens who are more respectful of their land. Colombia, Peru, and Chile are countries where very interesting STEM/STEAM education is being carried out, highlighting rural areas. At this point, we highlight Colombia as a country of great importance in Latin America in developing STEAM education. Colombia is clearly committed to STEM/STEAM education in rural areas.

We can observe that formal education advocates STEM education in Spain, since the LOMLOE Education Law contemplates it; this fact is of utmost importance. The main problem in Spain in rural areas is depopulation; although, the tendency to move to cities also affects other countries. Other factors must be taken into account here, not only educational factors but also the desired territorial model in each country. In Spain, there are no gender differences in education in rural areas as marked as they exist in the Latin American countries studied. In countries like Spain, the rural population is elderly, and this aging population in the villages can carry out engaging, informal education as they are a constant source of wisdom. It is also essential for older people to be informed and trained to stay up to date in today's world.

We conclude that education must be used to improve the rural environment and is vital for rural development. Students are trained in rural areas by valuing and knowing their resources, and their environment will have citizens who are more linked to it, thus awakening sensitivities from an early age, which will translate into a person–rural environment bond in different areas, such as cultural and social, environmental, emotional, professional, etc. STEM and STEAM education is necessary in our world, rural and urban, for students to develop and be better prepared for the needs of today's world. STEM and STEAM education presents interdisciplinary needs, in addition to training students to be more linked to the environment, the rural environment, and their land.

**Author Contributions:** E.G.-S., P.-T.E.-G., C.L.-B. and J.P.-H. were involved in conceptualization, research, formal analysis, and writing. E.G.-S., P.-T.E.-G., C.L.-B. and J.P.-H. have contributed to the development of this paper. The review and editing were performed by E.G.-S., P.-T.E.-G., C.L.-B. and J.P.-H. All authors have read and agreed to the published version of the manuscript.

**Funding:** This research was funded by funding entity Vice-rector for Research, Transfer and Dissemination Scientist at the National University of Distance Education (UNED).

**Data Availability Statement:** The original contributions presented in the study are included in the article, further inquiries can be directed to the corresponding author.

**Conflicts of Interest:** The authors declare no conflicts of interest.

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
