# Peer review of "Importance of STEM and STEAM Education for Improvement of the Land in the RURAL Environment: Examples in Latin America"

_land, doi:10.3390/land13030274_

Round 1

Reviewer 1 Report

Comments and Suggestions for Authors

Opening intro:

Interesting. I needed acronyms defined the first time the acronym is used, not later. I am not clear on the point.

Educating for the purpose of creating a population that is better environmental stewards is an admirable goal, but I don’t think this can be assumed to be a current universal goal. It possibly should, but this should not be presented as the current goal, but as something that should be aspired to globally – and may be required to sustain the planet.

Lines 191-198 – the 4 pillars felt out of the blue for me, maybe there is a commonly known reference I am unfamiliar with. “Gender” specifically feels like a tangential concept. These 4 pillars need more scaffolding to tie them in.

 The top of page 6 is a new author. Page 6-10 is interesting, but I am not sure of the purpose for this article.

I get lost in the listing on so many programs in so many countries. It would be more helpful to me to do a more in-depth case study of a program in a country, so we can learn what is and isn’t working and why.

The conclusion is very good. It ties the paper together well. I want a lot of this structure presented at the beginning. The article was very meandering to finally get to a point. All of the sections before the conclusion need to be tightened up. I would like more in-depth explanation of a particular program over listing. How do these programs address the 4 pillars presented in lines 190s? I am left wondering, what are the relevant themes?  The conclusion gives them to me but give them to me up front and then cut everything that does not serve these themes.

You also need to work on this shifting between authors, it varies greatly in style, this will be helped by making it more structured to the purpose of each section

Comments on the Quality of English Language

Line 368 repeated phrase

Line 622 typo – actually there are many many typos, I hope this is yet to be edited.

Author Response

Dear reviewer,

We want to thank your efforts in reviewing our article and for all your valuable comments.

Following your suggestions, we have made all the changes you suggested:

  1. We have defined the acronym STEM/STEAM the first time it is used.
  2. We have collected and incorporated the idea that educating to create a population that is a better guardian of the environment is an admirable objective but not a current global objective. Unfortunately, it must be aspired to at a global level to maintain the planet.
  3. The four pillars have changed by factors, and a table has also been added for better understanding. At the suggestion of explaining a program and seeing the four factors in it, it has been decided to explain Territorio STEM because several Latin American countries and several actors are involved within it and through a table it has been decided to unite with the four factors so important for the rural, clearly showing the reader the importance of working on them through STEM education precisely.
  4. This article strives to show the transformative potential of STEM/STEAM education in rural settings as an integrative force that drives the formation of new citizenships dedicated to social and environmental justice. In addition, it seeks to promote changes that foster a greater connection with the local territory, thus improving opportunities within this environment.
  5. In this article, we have compiled examples from Latin America where STEM and STEAM projects are being developed in which young people establish links with their environment, territory, and the surrounding environment. We must look carefully at the projects and works in which students' connection with their land and their environment is valued through the STEM and STEAM education necessary today. They show how this type of education has been widely researched in Latin American countries. Furthermore, Spain has also been investigated, not to make a precise comparison because it is impossible, but to highlight the countries' progress in improving the rural environment and their lands.
  6. Following your recommendations, the summary, introduction, objectives, and conclusion have been improved. In addition, a discussion section has been added. With all this, we consider the article more fluid and united.
  7. We have improved the bibliography, incorporating eight new references.
  8. The English of the article has been reviewed and improved.

Finally, we understand with the changes made thanks to your valuable advice, the article has improved significantly for publication.

Thank you again for your review and all your comments.

Our best wishes.

The authors

Reviewer 2 Report

Comments and Suggestions for Authors

Importance of STEM and STEAM education for improvement of the land in the RURAL Environment: examples in Latin America

Comments and Suggestions for Editor and Authors

This research endeavors to showcase the transformative potential of applying STEM/STEAM education in rural settings as an integrative force driving the formation of new citizenships dedicated to social and environmental justice. Simultaneously, it seeks to instigate changes that foster a stronger connection with the local territory, thereby enhancing opportunities within this environment. However, it is imperative that this objective be explicitly outlined in the Abstract.

While the introduction and literature review exhibit meticulous craftsmanship and maintain a high level of academic rigor, the articulation of the research objective lacks clarity throughout the text. To enhance reader comprehension, it is advised to explicitly state the research objective in the abstract, reiterate it at the conclusion of the introduction, and reintroduce it at the beginning of the conclusion, fostering a cohesive narrative.

The methodology and results analysis are presented with clarity and objectivity, reflecting a commendable standard of academic excellence. Nevertheless, there is a critical need for the study to draw broader conclusions. While some findings are presented, it is essential to underscore explicitly the societal impact that this research aims to achieve.

In summary, while the article exhibits strong academic skills in summarizing, introducing, and reviewing the literature, there exists an opportunity for improvement in clearly defining and emphasizing the research objective. Additionally, expanding upon the conclusions and contextualizing the anticipated social impact will contribute significantly to a more accessible and comprehensive interpretation of the text. Implementing these refinements will enhance the overall quality and impact of the research. The bibliography is extensive and current.

Comments on the Quality of English Language

In a scientific text, its quality is very important. The English in your writing seems fine to me, but I admit I'm not a native English speaker.

Author Response

Dear reviewer,

We want to thank your efforts in reviewing our article and for all your valuable comments.

Following your suggestions, we have made all the changes you suggested:

  1. The abstract has been modified as suggested.
  2. The objectives have been improved so that they are articulated with the full article's abstract and text. We presented this idea as you told us: abstract, end of introduction, and beginning of conclusion. In addition, a discussion section has been added.
  3. Conclusions have been improved and expanded, exposing the importance of STEM and STEAM education for the transformative potential of rural environments. With all this, the article is more fluid, united, and more coherent. Furthermore, the article has been presented as a source of reflection for teachers in rural areas, reflecting on the development of STEM and STEAM education in rural environments in our time.
  4. We think the article has been improved. In addition, information has been collected on the STEM Territory in several Latin American countries, uniting it with the four key factors for the rural environment compiled in a table.
  5. We have improved the bibliography, incorporating eight new references.
  6. The English has been reviewed and improved.

Finally, we understand with the changes made thanks to your valuable advice, the article has improved significantly for publication.The authors think that the article has been improved, and its reading can fulfill the reflection on the importance of STEM and STEAM education in rural environments to promote the training of new people, with greater social and environmental justice, with a greater connection with the local territory, thus improving the rural environment.

Thank you again for your review and all your comments.

Our best wishes.

The authors

Reviewer 3 Report

Comments and Suggestions for Authors

Dear Authors and Editors,

The study focuses on possible improvements in rural areas based on providing STEM/STEAM education. Thus, the paper proposes a description of selected country cases in Latin America and Spain and the characteristics of particular projects, approaches and employed instruments. However, based on the proposed study and its outcome, the reviewer cannot see scientifically grounded arguments that would substantiate that STEM/STEAM education is an integrative tool based on comparative analysis or other employed techniques in the study (see the objective in lines 144-148). Moreover, the construct in the study “…education for the improvement of the land in the rural environment…” and many expressions are not clearly explained as well. It is questionable whether the study employs a qualitative methodology that follows a particular structure and includes a clear enough discussion of the results. Instead, the study reminds “storytelling” that leaves behind many questions like “So what?”. The reader of the paper would require the evidence about provided education and its extent, content or organisation that promotes the improvements. More specifically:

1.     What do the authors mean by “improvement of the land” if considering different its dimensions, e.g. property, use, development, a value. Moreover, particularly in rural areas as this is the focus of the study, land issues can be studied if considering, e.g. soils, landscapes, land uses, land structures, land rights, etc.

2.     The clearly stated objective of the study/paper is not evident from the abstract. The long and unclear sentence appears in lines 16-20. The meaning of “a scenario” is not understandable and the abstract does not include the key outcome of the study there (in abstract) as well.

3.     The paper describes various concepts (e.g. new rurality) and instruments (e.g. blended learning), which is very good, however, those are related to different contexts and serve for the potential of improvements but not for new knowledge to be applied for improvements. So, the paper lacks a scientific discussion.

4.     Also, several other expressions are not clear to the reviewer and make it difficult to read and understand the manuscript, e.g. “…education is more developed in rural areas” - more compared with what? (countries, urban areas, etc., see lines 150-151); “…a comparison is made of various actions by Spain…”; “…institutional approaches” (see lines 174-175); “rural education” (see lines 180-181); unclear text in lines 188-191, etc.

5.     The objections of an editorial nature relate to Sections 4, 5 and 6. If there are no Sub-sections 4.2., 5.2. and 6.2. then it does not make sense to provide 4.1., 5.1. and 6.1.  

6.     Several assertions need references (for instance, see lines 567-572).

7.     The conclusions (Section 5 that should be a Section 9) are rather irrelevantly designed in both contexts of reflecting key results and scientific soundness of the paper.

8.     There are some other objections.

In view of all the above-mentioned, I recommend to the editors to REJECT the manuscript for publishing as it has serious flaws to be avoided.

Wishing you the best of luck with your further research.

With best regards

The reviewer

Author Response

Dear reviewer,

We want to thank your efforts in reviewing our article and for all your valuable comments.

Following your suggestions, we have made all the changes you suggested:

  1. The summary, introduction, objectives, and conclusions have been improved. In addition, a discussion section has been added so that the entire article is united for the reader.
  2. Reference has been made to the term soil in the article, explaining its importance and its connection with the rural environment, which is very important.
  3. Summary and objectives have been modified to be harmonious and united. A scenario has been removed.
  4. The authors do not consider that the article lacks scientific discussion. The research conducted in this article strives to show the transformative potential of applying STEM and STEAM education in rural settings as an integrative force that drives the formation of new citizenships dedicated to social and environmental justice. At the same time, it seeks to promote changes that foster a greater connection with the local territory, thus improving opportunities within this environment. The article has been enhanced to improve reader understanding, improving the summary, introduction, objectives, and conclusions. Furthermore, to facilitate the reader's understanding, important annotations of Territorio STEM that takes place in several Latin American countries have been collected in a table, and the authors have found in Territorio STEM the four factors that are so important for rural areas: education, health, technology, and employment.
  5. The authors think that the article has been improved, and its reading can fulfill the reflection on the importance of STEM and STEAM education in rural environments to promote the training of new people, with greater social and environmental justice, with a greater connection with the local territory, thus improving the rural environment.
  6. The sentence has been improved: education is more developed in rural areas. The text has been enhanced on lines 188-191, changing the four factors. Other concepts and ideas, such as rural education and institutional approaches, have remained the same because the authors of this article believe that they exist and are necessary for them to appear.
  7. Numbers 4.1., 4.2., 4.3 are eliminated.
  8. We have improved the bibliography, incorporating eight new references.
  9. The conclusions have been modified. The abstract, the introduction, the objectives, and the conclusions have been improved. Furthermore, we added discussions so that everything remains united and coherent.
  10. The English has been reviewed and improved.

The authors consider that the article has been improved and is research that aims to show the transformative potential of applying STEM and STEAM education in rural environments as an integrating force to train new people capable of being more socially and environmentally just while seeking time to foster a greater connection with the rural territory, thus improving the rural environment.

Finally, we understand with the changes made thanks to your valuable advice, the article has improved significantly for publication.

Thank you again for your review and all your comments.

Our best wishes.

The authors

Round 2

Reviewer 3 Report

Comments and Suggestions for Authors

Dear Authors and Editors,

The reviewer still cannot recognise the improved manuscript according to the previous review. In short, the additional scientific evidence is necessary. Moreover, the distinction between STEM and STEAM education has not been provided in the study.

Best wishes,

The reviewer

Author Response

Dear reviewer,

We want to thank your efforts in reviewing our article and for all your valuable comments.

Following your suggestions, we have made all the changes you suggested:

We think that a clear distinction is made in the manuscript between STEM and STEAM education when the latter incorporates the arts. In this latest version, the authors have collected some more scientific evidence added to the manuscript, such as Jiménez (2022) states, according to data from the Ibero-American Science Observatory, Technology and Society [OIE], 2020, educational management strategies must be reoriented to guarantee the increase of scientists, with interest in the formation of identities in science, technology, engineering, arts and mathematics, addressing global and national problems of Latin America and Caribbean countries and The STEM Women's Congress held in 2021 discussed how science unites with the environment through STEM education

Following your suggestions, we have made all the changes you suggested:

  1. The summary, introduction, objectives, and conclusions have been improved. In addition, a discussion section has been added so that the entire article is united for the reader.
  2. Reference has been made to the term soil in the article, explaining its importance and its connection with the rural environment, which is very important.
  3. Summary and objectives have been modified to be harmonious and united. A scenario has been removed.
  4. The authors do not consider that the article lacks scientific discussion. The research conducted in this article strives to show the transformative potential of applying STEM and STEAM education in rural settings as an integrative force that drives the formation of new citizenships dedicated to social and environmental justice. At the same time, it seeks to promote changes that foster a greater connection with the local territory, thus improving opportunities within this environment. The article has been enhanced to improve reader understanding, improving the summary, introduction, objectives, and conclusions. Furthermore, to facilitate the reader's understanding, important annotations of Territorio STEM that takes place in several Latin American countries have been collected in a table, and the authors have found in Territorio STEM the four factors that are so important for rural areas: education, health, technology, and employment.
  5. The authors think that the article has been improved, and its reading can fulfill the reflection on the importance of STEM and STEAM education in rural environments to promote the training of new people, with greater social and environmental justice, with a greater connection with the local territory, thus improving the rural environment.
  6. The sentence has been improved: education is more developed in rural areas. The text has been enhanced on lines 188-191, changing the four factors. Other concepts and ideas, such as rural education and institutional approaches, have remained the same because the authors of this article believe that they exist and are necessary for them to appear.
  7. Numbers 4.1., 4.2., 4.3 are eliminated.
  8. We have improved the bibliography, incorporating eight new references.
  9. The conclusions have been modified. The abstract, the introduction, the objectives, and the conclusions have been improved. Furthermore, we added discussions so that everything remains united and coherent.
  10. The English has been reviewed and improved.

The authors consider that the article has been improved and is research that aims to show the transformative potential of applying STEM and STEAM education in rural environments as an integrating force to train new people capable of being more socially and environmentally just while seeking time to foster a greater connection with the rural territory, thus improving the rural environment.

Finally, we understand with the changes made thanks to your valuable advice, the article has improved significantly for publication.

Thank you again for your review and all your comments.

Our best wishes.

The authors
